# Improving Human Legibility in Collaborative Robot Tasks through Augmented Reality and Workspace Preparation

Yi-Shiuan Tung
yi-shiuan.tung@colorado.edu
University of Colorado Boulder
Boulder, Colorado, USA

Matthew B. Luebbers
matthew.luebbers@colorado.edu
University of Colorado Boulder
Boulder, Colorado, USA

Alessandro Roncone
alessandro.roncone@colorado.edu
University of Colorado Boulder
Boulder, Colorado, USA

Bradley Hayes
bradley.hayes@colorado.edu
University of Colorado Boulder
Boulder, Colorado, USA

## ABSTRACT

Understanding the intentions of human teammates is critical for safe and effective human-robot interaction. The canonical approach for human-aware robot motion planning is to first predict the human's goal or path, and then construct a robot plan that avoids collision with the human. This method can generate unsafe interactions if the human model and subsequent predictions are inaccurate. In this work, we present an algorithmic approach for both arranging the configuration of objects in a shared human-robot workspace, and projecting "virtual obstacles" in augmented reality, optimizing for legibility in a given task. These changes to the workspace result in more legible human behavior, improving robot predictions of human goals, thereby improving task fluency and safety. To evaluate our approach, we propose two user studies involving a collaborative tabletop task with a manipulator robot, and a warehouse navigation task with a mobile robot.

## ACM Reference Format:
Yi-Shiuan Tung, Matthew B. Luebbers, Alessandro Roncone, and Bradley Hayes. 2023. Improving Human Legibility in Collaborative Robot Tasks through Augmented Reality and Workspace Preparation. In *Proceedings of 6th International Workshop on Virtual, Augmented, and Mixed-Reality for Human-Robot Interactions (VAM-HRI '23).* ACM, New York, NY, USA, 5 pages.

## 1 INTRODUCTION

In human-robot collaborative tasks, shared mental models between agents enable the awareness and joint understanding required for effective teamwork. Notably, the synchronization of mental models allows human and robot teammates to collaborate fluently, adapt to one another, and build trust [22]. With no shared notion of the task to be completed, the inherent unpredictability and opacity of human decision-making makes robot planning difficult [21]. To this end, prior research efforts have focused on developing methods for robots to express their intentions to human teammates, as well

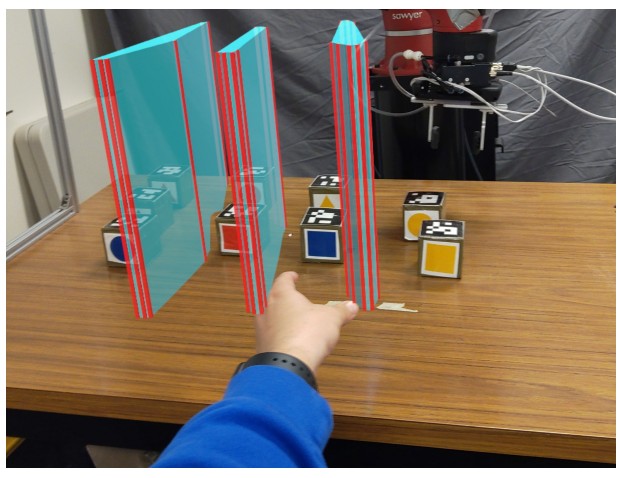

**Figure 1: Legible workspace configuration for a tabletop collaborative task as viewed through the HoloLens AR interface. By rearranging items and projecting virtual obstacles (shown in cyan with red edges), the robot improves its probability of predicting the correct cube the human is reaching for (in this case, the blue square cube).**

as methods for modeling and understanding human teammates' intentions.

For robots to express their intentions through motion planning, Dragan et al. [4] formalized the notion of legibility: the probability of successfully predicting an agent's goal given an observation of a snippet of its trajectory. Other research efforts have focused on developing robots that can predict human behavior [19], generating motion plans to safely interact in a shared environment. However, these methods are limited by how well the robot is able to predict a human collaborator's intention and resultant behavior. With inaccurate human models or unexpected human behavior diverging from past experiences, the robot may produce unsafe interactions [14].

In this work, we improve the robot's ability to predict the human's goal (i.e. human legibility) by rearranging items in the shared workspace and projecting "virtual obstacles" in augmented reality (AR). We present a legibility metric that scores potential workspace configurations in terms of how legible the actions of a human

teammate are likely to be when performing a task. Each candidate workspace configuration combines a potential arrangement of physical and virtual objects in the environment. Virtual obstacles are used to impose constraints on the possible motions of the human, further forcing them to move legibly when approaching a goal. Moreover, AR-based virtual obstacles are able to be reconfigured rapidly as the task progresses, removing the rigidity imposed by a fixed legibility-optimized workspace configuration for the entire duration of a task. The AR interface ensures that legible human trajectories can continually be elicited given the current task context.

We efficiently explore the space of workspace configuration solutions using a quality diversity algorithm called Multi-dimensional Archive of Phenotypic Elites, or MAP-Elites [15]. Instead of finding a single optimal solution, MAP-Elites produces a map of high-performing solutions along dimensions of a feature space chosen by the designer. MAP-Elites enables efficient and extensive exploration of complex search spaces, such as the placement of workspace objects and virtual obstacles in continuous space, leading to higher quality solutions compared with other search algorithms. In this work, we describe three primary contributions: 1) an algorithm for optimizing the physical arrangement of a shared human-robot workspace for human legibility, 2) an AR interface for projecting dynamically generated virtual obstacles into the workspace to further improve human legibility, and 3) proposed experiments to evaluate our combined approach in tabletop and navigation human-robot collaborative tasks.

## 2 RELATED WORK

*Modeling Human Motion:* Avoiding collisions with human collaborators is an important problem in human-robot collaboration, which is often solved by attempting to model human goals, and resultant trajectories. Lasota et al. developed a human-aware motion planning algorithm that approximates the segments of the workspace a human might occupy and modifies the robot's plan to avoid possible collisions [10]. Other works have used Partially Observable Markov Decision Processes to determine optimal actions for a robot while maintaining a probabilistic belief over a human's intended goals [8, 17]. Our work focuses on reducing the uncertainty inherent in modeling the intentions of human collaborators by pushing them towards legible behavior.

*Environment Modification in Robotics:* Prior work has explored robots deliberately modifying their environment in order to better achieve their goals across a variety of domains. Fujisawa et al. developed a robot that can construct auxiliary structures to facilitate its own movement across unknown rough terrain [7]. Shao et al. provided a robot with environmental fixtures that imposed constraints on its range of motion, enabling more robust manipulation [20]. Another work maximized the coverage area of a floor cleaning robot by providing automated suggestions for the optimal placement of objects and furniture [16]. These works show the potential advantages of a robot modifying its environment to improve task performance. In our work, the robot modifies its environment with the explicit goal of improving its ability to collaborate with a human teammate.

*Improving Robot Legibility:* Legible robot motion, where the robot's goal is easily inferred by human teammates, produces desirable interactions when robots work in human environments. Robots with highly legible arm movements lead to greater task efficiency, trustworthiness, and sense of safety [3]. Legible robot navigation in mobile robots results in fewer stops due to potential collisions [2]. Another work found that human teammates prefer legible task allocation, where they have a clear idea of what each agent's role is [9]. The benefits of legibility have also been explored in sequential decision making [5] and robot to human handovers [11]. While prior work has extensively studied producing legible robot motion, our work's focus on improving the legibility of human teammates is novel.

*Augmented Reality for Robotics:* Augmented reality (AR) interfaces are useful for compactly conveying a robot's intentions to human collaborators. Walker et al. used AR to visually communicate the flight paths of drones [23]. Rosen et al. displayed projections of future robot arm motion to communicate intent [18]. AR interfaces can also visually represent the tasks the robot will execute in order to improve workspace safety [1] or assist with the debugging of those tasks [12]. Our work utilizes an AR interface to display virtual, configurable obstacles which elicit legible motions from a human teammate.

## 3 APPROACH

In this section, we describe our approach for modifying the shared human-robot workspace to maximize legibility. We first introduce the legibility score used to evaluate a specific workspace configuration (Sec. 3.1), followed by the optimization framework used to maximize legibility for a task (Sec. 3.2). We then demonstrate how to find an approximate solution to the optimization using MAP-Elites (Sec. 3.3). Lastly, we discuss our augmented reality interface for visualizing virtual obstacles that further improve the human's legibility (Sec. 3.4).

### 3.1 Legibility Metric

To evaluate the legibility of a given workspace configuration, we consider the probability distribution of predicting that the human is approaching goal $G$ given an observed human trajectory from $S$ to $Q$. We use the formulation developed by Dragan et al. [4] shown in Equation 1.

$$\Pr(G|\xi_{S \to Q}) \propto \frac{exp(-C(\xi_{S \to Q}) - C(\xi^*_{Q \to G})}{exp(-C(\xi^*_{S \to G}))} \qquad (1)$$

The optimal human trajectory from $X$ to $Y$ with respect to cost function $C$ is denoted by $\xi^*_{X \to Y}$. Equation 1 evaluates how efficient (with respect to $C$) going to goal $G$ is given the observed trajectory $\xi_{S \to Q}$ relative to the most efficient trajectory $\xi^*_{S \to G}$.

Let $\mathcal{G}$ be the set of valid goals at the current time step. We develop a legibility score (Equation 3) for use in our optimization objective that, for every valid goal at a given time step in the task execution, maximizes the margin of prediction between the human's chosen goal $G_{true} \in \mathcal{G}$ and all other valid goals. If the most likely goal is not $G_{true}$, the score is penalized by a fixed cost $c$. Otherwise, the score is the difference of the two highest probabilities shown in Equation 2. The notation $G_{(i)} \in \mathcal{G}$ denotes the $i$th index of a sorted

list constructed from $\mathcal{G}$, ordered from smallest likelihood to largest given the observed trajectory ($\xi_{S \to Q}$).

$$margin(\mathcal{G}|\xi_{S \to Q}) = G_{(n)} - G_{(n-1)} \qquad (2)$$

$$\text{EnvLegibility}(G_{true}) = \begin{cases} -c, & \text{if } \arg\max_{G \in \mathcal{G}} \Pr(G|\xi_{S \to Q}) \neq G_{true} \\ margin(\mathcal{G}|\xi_{S \to Q}), & \text{otherwise} \end{cases}$$
$$(3)$$

## 3.2 Optimization for Task Legibility

We use the structure of the task to find the set of valid goals $\mathcal{G}$ at any given time step. Instead of considering that the human is approaching all the possible goals in the workspace, the precedence constraints in the task $T$ reduces that to a subset of the goals.

To formalize this, let $T$ be a task that consists of $m$ subtasks $t_1...t_m$. There exists precedence constraints denoted by $t_i \to t_j$ so that subtask $t_i$ has to be completed before subtask $t_j$ can begin. To generate a workspace configuration with improved legibility of the agent's goals for the task $T$, the objective function is to maximize the legibility metric from Equation 3 for all valid sequences of $T$ (Equation 4). When the human is working on subtask $t$ in the task sequence $T$, $\mathcal{G}$ is the set of goals corresponding to subtasks that have all precedence constraints satisfied.

$$\max \sum_{T' \in \text{permutations}(T)} \mathbb{1}\{\text{valid}(T')\} \times \sum_{t \in T'} \sum_{G \in \mathcal{G}} \text{EnvLegibility}(G)$$
$$(4)$$

The $\mathbb{1}\{f\}$ is an indicator function that returns 1 if the function $f$ is true and 0 otherwise. Equation 4 is the objective function when optimizing for legible workspace configurations.

## 3.3 Search using Quality Diversity

Iterating through all possible workspace configurations to find the optimal solution is computationally intractable for most applications. The number of possible configurations is exponential in the number of goals, virtual obstacles, and size of the workspace. We use MAP-Elites [15] to approximate the optimal solution.

Algorithm 1 shows the pseudocode for MAP-Elites. The algorithm takes as input a function $G_H$ that outputs human trajectory given a goal in the workspace. $G_H$ can be learned from data via inverse optimal control [13] or approximated via shortest path to goal [4]. In addition, the user chooses an objective function $F$ to evaluate each solution (in our case, the legibility objective defined in Equation 4) and a set of features determined by a measure function $M$. Given a candidate solution, the measure function outputs a set of features that are the dimensions of solution map $S$. For example, in our proposed tabletop experiment, the features are the minimum distance between the cubes and the ordering of the cubes from left to right and top to bottom. The cube ordering for Figure 2b is red square, yellow square, blue square, red circle and so on.

MAP-Elites consists of two phases: initialization and improvement. In the initialization phase, we randomly sample workspaces (Line 3) and store them in the cell that they belong to according to their features (Lines 7-11). In the improvement phase, we follow [6] to use gradient information to speed up search. We first randomly sample from the map of solutions (Line 5) and then run gradient descent to improve the solution (Algorithm 2). In Algorithm 2, Line 1

retrieves all available perturbations to the workspace (i.e. changing an item's position, adding or removing a virtual obstacle). For each perturbation, a new workspace configuration $w'$ is generated by applying the perturbation. If the legibility score of $w'$ is better than the current best workspace $w^*$, then $w^*$ is updated to $w'$ (Lines 2-7). If there was an improvement to the workspace, we run gradient descent again (Lines 8-9). Otherwise, a local minima has been found, and we return the best workspace found (Line 11).

The perturbations for the available_perturbations function on Line 1 are obtained as follows: for each item in the workspace, we sample from a Gaussian centered at the current $x$ and $y$ position with some variance. Furthermore, we sample random locations for the placement of fixed-size virtual obstacles. If the virtual obstacles overlap, we combine them by taking the convex hull of the obstacles.

---

**Algorithm 1:** Workspace Generation with MAP-Elites

**Input:** Human Trajectory Generator $G_H$, Objective function $F$, measure function $M$

**Initialize:** Solution map $S \leftarrow \emptyset$, Solution values $V \leftarrow \emptyset$

1 **for** $i = 1, ..., N$ **do**
2    **if** $i < N_{init}$ **then**
3       Generate workspace $w$ = random_workspace()
4    **else**
5       Select random workspace from map $w$ = random(S)
6       Run $w$ = improve_via_gradient_descent($w$)
7    Determine features $\boldsymbol{m} = M(w)$
8    Determine objective score $s = F(w)$
9    **if** $S[\boldsymbol{m}] = \emptyset$ *or* $s < V[\boldsymbol{m}]$ **then**
10       $S[\boldsymbol{m}] = w$
11       $V[\boldsymbol{m}] = s$
12 **return** S, V

---

**Algorithm 2:** improve_via_gradient_descent

**Input:** Workspace configuration $w$, objective score of w $s_w$, objective function $F$, measure function $M$

**Initialize:** Best configuration $w^* \leftarrow w$, Best score $s^* = s_w$

1 Get perturbations $A$ = available_perturbations($w$)
2 **for** $a \in A$ **do**
3    New workspace $w'$ = apply_perturbation($w, a$)
4    Compute legibility score $s_{w'} = F(w')$
5    **if** $s_{w'} \leq s^*$ **then**
6       $w = w'$
7       $s^* = s_w$
8 **if** $w^* \neq w$ **then**
9    **return** improve_via_gradient_descent($w^*$)
10 **else**
11    **return** $w^*$

---

## 3.4 AR Interface

The boundaries of the algorithmically generated virtual obstacles are passed to an augmented reality interface, implemented using

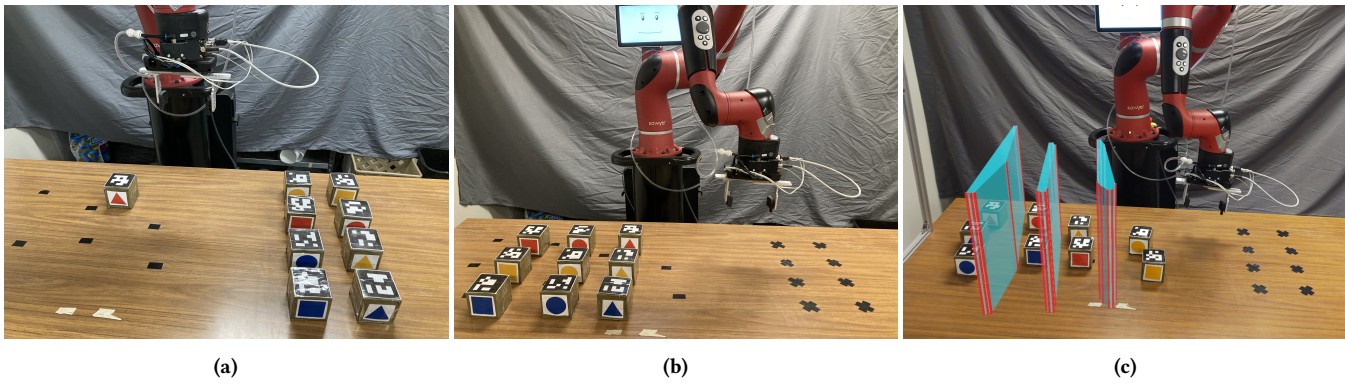

**Figure 2: Tabletop task with the Sawyer robot. (a) The experiment involves the human and the robot taking turns to move a set of cubes to a desired goal configuration. (b) Initial setup for the baseline condition where the cubes are organized by their color. (c) Legibility-optimized setup as viewed from the HoloLens AR interface, showing the algorithmically generated arrangement of cubes along with the generated virtual obstacles. The workspace setup is designed to improve the probability of the robot correctly inferring the human's goal.**

a Microsoft HoloLens 2 head-mounted display. The AR interface renders those obstacles directly in the environmental context of the shared workspace as holograms of cyan barriers with red outlines (Fig. 1 and Fig. 3). These barriers appear to the human as if they are physically located in the environment, and indicate regions of the environment the human should not enter. Through their parameterization, the location of the barriers force humans into highly legible patterns of movement.

## 4 EXPERIMENTS

To evaluate our approach, we propose human subjects studies in two separate domains: tabletop manipulation and warehouse navigation. These domains evaluate different aspects of the legibility of human motion (arm reaching for tabletop manipulation and walking for navigation), demonstrating the potential benefits of our approach across varied human robot interaction settings.

### 4.1 Tabletop Task with Manipulator Robot

This domain is representative of tasks where human and manipulator robot teammates work together in the same tabletop workspace, such as furniture assembly and fruit packaging. The goal of each experimental task is to move a set of cubes into a desired end configuration (Fig. 2a). The human-robot interaction will follow a leader-follower, turn-taking paradigm, with the human placing the first cube, and the robot placing the second, alternating until the task is finished. At the start of each turn, the robot maintains a probability distribution over the possible cubes the human is reaching for in real time. Once the robot is sufficiently confident of the human's goal, the robot will select its own cube to pick up and move to grasp it. The precedence constraints are set such that the first column of the desired configuration must be completed before the second column can start.

We will compare our proposed legibility-optimized workspace configuration against a baseline where cubes of the same color are placed near each other. Fig. 2b shows the initial setup for the

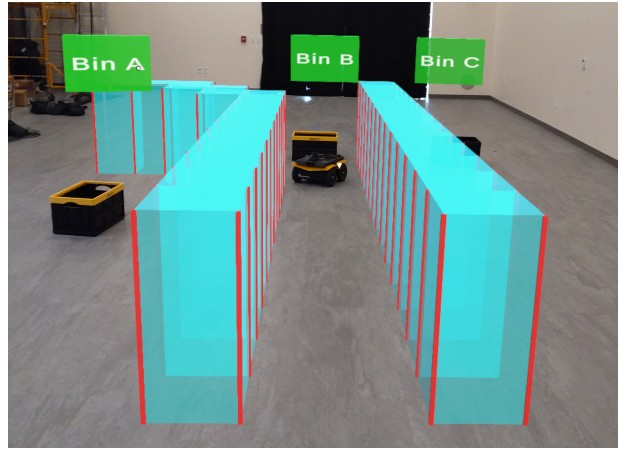

**Figure 3: Legible workspace configuration for the warehouse navigation task as viewed from the HoloLens AR interface. For their next part, the human can either go to Bin A, Bin B, or Bin C. The virtual obstacles are placed such that it is immediately clear which bin the human is heading towards after their first few steps. The mobile robot teammate is visible in front of Bin B.**

baseline condition, while Fig. 2c shows the legible setup (with AR obstacles included). We hypothesize the following:

- **H1:** The robot will be able to predict the human's goal faster in a legibility-optimized environment as compared to the baseline.
- **H2:** The human participant will rate the collaboration as more fluent and will prefer working with the robot in a legibility-optimized environment.

## 4.2 Warehouse Navigation with Mobile Robot

This domain is representative of tasks where human and mobile robot teammates work together in the same large open floor workspace, such as warehouse stocking. The human participant will be tasked with moving specified parts from specified bins to a target repository for packaging and shipping. Meanwhile, a mobile robot will conduct inventory checks on the same set of bins. To avoid collisions where the mobile robot and human are attempting to visit the same bin at the same time, the robot maintains a probabilistic model of the bin the human is intending to approach, choosing a goal and generating a collision-free path in response to that model. The robot will stop to replan if the belief in the human's most likely goal changes. In the legibility-optimized condition, participants will view dynamic virtual obstacles projected onto the floor via an AR headset, attempting to make their motions as legible as possible as the task progresses.

We compare our legibility-optimized approach with a baseline where no virtual obstacles are displayed in AR. Our hypotheses are as follows:

- **H3:** The robot will replan fewer times and will accomplish its tasks faster in the presence of dynamic AR obstacles as compared to the baseline.
- **H4:** The human participant will rate the collaboration as safer and will prefer working with the robot in the presence of dynamic AR obstacles.

## 5 CONCLUSION

In this work, we proposed an approach to improve human teammate legibility during human-robot collaboration, by strategically rearranging the shared workspace and projecting virtual obstacles via AR. We hypothesize that these legibility-optimized environment setups will improve joint task fluency and safety, as well as subjective ratings of the robot. To evaluate those claims, we also proposed a pair of human subjects experiments to validate our approach across two distinct human-robot collaboration domains.

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
