# OpenReview forum: "Improving Human Legibility in Collaborative Robot Tasks through Augmented Reality and Workspace Preparation"
_humanrobotinteraction.org/HRI/2023/Workshop/VAM-HRI — VAM-HRI 2023 Oral_

### Official Review · Program_Chairs · 2023-02-24
**Great and innovative work**

**Rating:** 9
**Confidence:** 5

**Review:**

Reviewer 1

In this work authors present an AR interface for visualizing virtual obstacles in HoloLens interface, improving human legibility and human-robot understanding of different tasks. This can be successfully communicated and improve the users' interaction with the robot, making it easier and clearer for the user what is the goal of the task. Additionally, the obstacles/no-go zones can be updated while the task progresses. The paper shows a great example of how the robot can better communicate its intentions to improve human understanding, effectively not blocking human’s movements but rather directing subtly the humans.

Potential improvements.

It would be interesting to see the results of a couple of participants.

The method sounds very promising but what do you compare it with? Is the baseline completing the same task but without the virtual obstacles?

You talk about the dynamic update of the obstacles, how fast can it be done? I.e. if the movements of the human are too fast will the perception module of the robot succeed to update everything in a reasonable time?

What happens if human don’t follow robot’s suggestions and crosses the obstacles? Will the robot manage to update the legible workspace and continue with the task or it may fail? I think it would be interesting to write/discuss such edge cases.


------------------------------


Reviewer 2

This paper presents a method to improve human motion legibility through the use of AR barriers the signal to users where they should or should not move. This method hypothesizes that the virtual barriers will improve robot predictions of human future movement to improve task efficiency and safety. The submission details a legibility score that is generated by the system as well as a method to optimize for task legibility. This allows the AR system to algorithmically generate the virtual objects that present user restrictions during a collaborative and/or collocated task with a human. A series of experiments are presented for future work. The novel system presented in this paper is highly relevant to this venue and I recommend its acceptance.

Questions and Comments:
- What is the basis for the design choice of making the barriers cyan with red borders? Further information and/or justification behind these (and any other future) design choices would strengthen this paper.
- I am very interested in seeing how this system performs when users disregard the AR barriers.
- “The human-robot interaction will follow a leader-follower, turn-taking paradigm, with the human placing the first cube, and the robot placing the second, alternating until the task is finished. At the start of each turn, the robot maintains a probability distribution over the possible cubes the human is reaching for in real time. Once the robot is sufficiently confident of the human’s goal, the robot will select its own cube to pick up and move to grasp it” I think this sub section could use some additional clarification, as this does not seem like a pure alternating/turn-based task like it initially sounds like and more like the robot starts moving while the human is still moving (unless I am mistaken).
- This paper could be potentially strengthened by comparing multiple different design choices for the AR condition (e.g., walls with different textures such as fire or electrified or sharp, walls that animate, walls with classic warning signs, walls that respond to human collisions to reinforce the notion they should not be touched/crossed, walls that change color intensity based on human proximity, etc.)

---

### Decision · Program_Chairs · 2023-03-02

Accept (Oral)